# Influence of Forest Visitors’ Perceived Restorativeness on Social–Psychological Stress

**DOI:** 10.3390/ijerph18126328

**Published:** 2021-06-11

**Authors:** Don-Gak Lee, Mi-Mi Lee, Young-Mi Jeong, Jin-Gun Kim, Yung-Kyoon Yoon, Won-Sop Shin

**Affiliations:** 1Graduated Department of Forest Therapy, Chungbuk National University, Cheongju 28644, Korea; don0810@naver.com (D.-G.L.); yulim741@naver.com (M.-M.L.); cozy1018@naver.com (Y.-M.J.); 2Department of Forest Sciences, Chungbuk National University, Cheongju 28644, Korea; yyk5757@hanmail.net

**Keywords:** COVID-19, social–psychological stress, PWI-SF (Psychosocial Well-being Index Short Form), ART (attention restoration theory), PRS (Perceived Restorativeness Scale), forest cultural and recreational resources

## Abstract

This study was conducted to verify the perceived restorativeness of citizens visiting forests on social–psychological stress and psychological resilience according to forest space type. The study involved a questionnaire survey conducted on citizens who visited forests between 1 May and 15 July 2020, when social distancing in daily life was being implemented. Three types of forest spaces (urban forest, national park, and natural recreation forest) were selected for the survey. They used the survey results of 1196 people as analysis data for this study. In this study, the PRS (Perceived Restorativeness Scale) and the PWI-SF (Psychosocial Well-being Index Short Form) were used to evaluate perceived restorativeness and social–psychological stress of citizens visiting forests. In the study, the average score of visitors’ perceived restorativeness was 5.31 ± 0.77. Social–psychological stress was found in the healthy group, potential stress group, and high-risk group. These groups made up 8.0%, 82.5%, and 9.5% of the respondents, respectively. Pearson’s correlation analysis between perceived restorativeness and social–psychological stress revealed that the higher the perceived restorativeness, the lower the social–psychological stress. “Diversion Mood”, “Not bored”, and “Coherence”, which are the sub-factors of perceived restorativeness according to the forest space type, were found to have meaningful results for psychological resilience. However, there was no significant difference in the forest space type between “Compatibility” and social–psychological stress, which are sub-factors of perceived restorativeness. In conclusion, the forest space type affects the psychological resilience of those who visit the forest. Urban forests, national parks, and natural recreation forests are places to reduce stress.

## 1. Introduction

Unprecedented circumstances such as urbanized society, sedentary lifestyle, unstable employment, climate change, and COVID-19 have led to an increase in stress and anxiety across the world’s population [1,2,3]. In particular, the COVID-19 epidemic, a new disease that causes severe acute respiratory syndrome caused by coronavirus (SARS-Cov-2), affects our society as a whole, individuals, and families, and also composes multidimensional stressors including physical risk, disruption in daily life, uncertainty, social isolation, economic loss, and unemployment risk [4,5]. Stress brings a significant increase in noncommunicable diseases such as cardiovascular, metabolism, immunology, oncology and psychiatric disorders [6,7]. Noncommunicable diseases can be prevented through healthier environments, but the incidence and mortality rates are increasing worldwide [8]. Therefore, measures to recover from stress and promote healthy lifestyles are very important to public health. The natural environment gives a sense of psychological satisfaction and stability, reduces stress, and increases immunity [9]. The nature of the forest environment itself can positively act on the psychology of the injured individual [10], and activities in the forest can have a positive effect on human stability and are particularly effective in enhancing the happiness of infants [11,12]. In addition, green spaces for the elderly are effective in health promotion, giving physiological and psychological effects and expanding social networks [13]. Nature encountered in daily life can relieve stress and fatigue [14]. Being in contact with the natural environment for 15 min not only lowers the cortisol concentration that rises in a stressed state [15], but also increases human immunity, and just walking slowly in the forest relieves stress and brings mental stability [16].

The healing effect in the natural environment can be related to the recovery environment, and there is a positive correlation between the natural environment and the recovery environment [17,18,19]. The more exposed to the recovery environment, the better the mood, and this alleviates daily problems and minor stresses [20]. Kaplan and Kaplan [21,22] argued for the psychological benefits of the natural environment as Attention Restoration Theory (ART). In the urban environment in which we live, there are many factors that require attention, such as automobiles, noise, neon signs, and air pollution, and in order to live daily life, we need to pay attention to changes in the surrounding environment. An environment that helps us overcome fatigue is called a restorative environment. For people tired of modern urban life, the natural environment of the forest is a representative recovery environment, and stress is reduced through an environment that does not require attention. The recovery of effective functioning is enabled by settings that have specific vital properties, such as “being away”, “extent”, “fascination”, and “comparability”. These components refer to those critical properties of forests that trigger psychological states, contributing to the therapeutic experience [21,22].

In the state recovery theory, the representative environment that has a lot of restorative environmental factors is considered to be the natural environment, and by contacting the natural environment, attentional ability is restored, and it can effectively suppress competition and distraction stimuli arising from the external or internal environment. As a result, the likelihood of negative emotions is lowered [21]. When attention is restored by contact with the natural environment, it has an emotional boosting effect by mitigating the negative effects of stress [23]. Therefore, it is necessary to develop resilience to prevent stress and mental disorders caused by COVID-19 [24]. It is important to study the perception of the recovery environment of forests for the Korean people who are under daily stress due to concerns about virus infection.

Forest leisure activities refer to various activities based on green areas such as forests and the use of forests to relieve mental and physical stress, self-charge, and rest in individual free time [25]. Forest culture and recreation in Korea includes all forest resources that are used for rest and healing of mind and body through the interaction of forests and humans, and forest space types include national parks, natural parks, urban forests, natural recreational forests, and living green spaces [26]. The activities that enhance the human body’s immunity and promote health by utilizing various elements of nature such as the fragrance of the forest and scenery are defined as forest healing (Article 2, No. 4 of the Forest Culture and Recreation Act). Forest healing activities have the effect of improving mental health by improving negative emotions such as anxiety and depression and have an emotional stability effect that induces positive psychological symptoms by improving the quality of life of individuals such as subjective and psychological well-being and happiness [27].

The campaign to prevent the spread of COVID-19 at the government level is prolonged, and although the National Park Management Corporation restricts the use of facilities, the number of visitors to forests near the city center, who are easily accessible by private vehicles, is increasing [28]. Like Wilson’s [29] biophilia hypothesis that humans are instinctively attracted to forests because entering a forest lowers stress, will the stress of those who visit forests decrease? As Ulrich’s [30] study shows that stressed people have positive influences such as friendship, affection, and joy from natural environmental factors, will people who visit forests increase their perceived restorativeness?

In this study, we tried to find out how the perceived restorativeness of forest visitors affects social–psychological stress. The research hypotheses are as follows:(1)The perceived restorativeness will have a significant influence on the social–psychological stress of citizens.(2)Depending on the type of forest space, the perceived restorativeness will affect the psychological sense of recovery.

## 2. Materials and Methods

### 2.1. Study Subject and Research Method

This study was conducted in compliance with the government’s policy to maintain distance in daily life for COVID-19 by selecting three survey sites including Mt. Gwanaksan Urban Forest (hereinafter referred to as Urban Forest), Mt. Bukhansan National Park (hereinafter referred to as National Park), and Mt. Yumyeongsan Natural Recreation Forest (hereinafter referred to as Natural Recreation Forest) during the period from 1 May to 15 July 2020. A questionnaire was conducted with 19-year-old adults who visited the forest. As part of social distancing, the researchers wearing masks explained the contents of the questionnaire to the descending people at the midpoint of each place. They conducted the questionnaire after obtaining oral consent. This study prevented people from gathering in groups by sending an official letter to each site management office and requesting cooperation. A total of 1196 questionnaire responses were adopted for the data of this study, and out of a total of 1201 questionnaires, 5 questionnaires that were not completely checked were excluded.

### 2.2. Study Scope

Forest culture and recreation in Korea include all forest resources used for mental and physical relaxation and healing through the interaction of forests and humans. Forest space types include national parks, natural parks, urban forests, natural recreational forests, and living green spaces [26]. This study selected urban forests, national parks, and natural recreational forests in the suburbs of Seoul as the target sites to find out visitors’ perceived restorativeness according to the forest space type (Figure 1; Table 1).

The Gwanak urban forest is a neighborhood park with good accessibility, as it is adjacent to a residential area with a subway station, highway toll gate, and bus stop. The size of the Gwanak urban forest is 19,220,000 square meters and was established in 1968. The main species are *Pinues densiflora*, *Alnus hirsuta*, *Quercus*, *Fraxinus*, *Fallopia convolvulus*, and *Buxus*. The main facilities are trails with a total of 32.

The Mt. Bukhan national park is a rare natural park in the city center globally, and it fully preserves the beautiful natural scenery and provides a pleasant exploration service. It is designated as the 15th national park in Korea. The size of the forest is 76,922,000 square meters. The Mt. Bukhan national park was established in 1983, and mature trees have shaped the forest landscape. The main species are *P. densiflora*, *A. hirsute*, *Quercus mongolica*, *Juniperus chinensis*, *Actinidia polygama*, and *P. bungeana*. It contains over 1300 species of animals and plants that inhabit the beautiful natural environment.

The Mt. Yumyeong recreation forest is a beautiful place with a gentle and soft ridge-line, a valley with abundant water, and strange rocks and lush forests. The size of the Mt. Yumyeong recreation forest is 89,200,000 square meters. The main species are *Q. mongolica*, *Acer palmatum*, *Styrax obassia*, *Pinus koraiensis*, and *Larix leptolepis*. The main facilities are circular trails (6 km), physical training facilities, a log cabin, toilets, barbecue grounds, and a car campground.

### 2.3. Measurements

#### 2.3.1. The Perceived Restorativeness Scale (PRS)

Hartig et al. [31] saw that the recovery environment could help recover depleted psychology. The PRS was used to develop a scale to measure the psychological factors considered effective in the recovery experience. The recovery environment perception scale of this study is divided into four categories of 16 items in total. The sub-factors are Diversion mood, Not boring, Coherence, and Compatibility [32,33]. The Likert-7 scale was renamed and used. In this study, the coefficient of item internal consistency was found to be 0.86 for diversion mood, 0.66 for not boring, 0.60 for coherence, and 0.87 for suitability, and the overall perceived restorativeness was 0.87 for each sub-factor.

#### 2.3.2. Psycho Social Well-Being Index Short Form (PWI-SF)

The social–psychological stress PWI-SF is the Likert-4 scale of “not at all” to “always” of the degree of the physical and psychological state experienced or felt in recent weeks [34]. PWI-SF converts the score to 0–1–2–3, and the total score is distributed between 0 and 54 points. Thus, 0–8 points are defined as a healthy group, 9–26 points a potential stress group, and 27–54 points as a high-risk group, meaning that the higher the score, the higher the stress level [35]. In this study, based on the classification of PWI-SF [34], participants were divided into a healthy group, a potential stress group, and a high-risk group, and the in-item fit coefficient was found to be 0.88.

### 2.4. Data Analysis

The G*power 3.1 program (Heinrich Hein University, Düsseldorf, Germany) was used to calculate the number of subjects required for this study. The effect size was set to 0.03, the power was 95%, the significance level was 0.05, the number of factors was 2, and the calculation result was calculated using multiple linear regression analysis. As a result, a total of 518 subjects were calculated. The data of 385 urban forest visitors (32.2%), 493 national park visitors (41.2%), and 318 natural recreational forest visitors (26.6%), and a total of 1196 people were collected in this study and were suitable for the number of samples, so the SPSS 25.0 program was used. For data analysis, first, frequency analysis on demographic characteristics, calculation of the inter-item fit coefficient to confirm the reliability of each scale, descriptive statistics on major variables, and Pearson correlation analysis were performed. Second, a chi-square test was conducted to verify the difference in social–psychological stress levels according to demographic characteristics. Third, a one-way variance analysis was conducted to verify the difference in significant variables according to the forest space type.

## 3. Results

### 3.1. Demographic and Sociological Characteristics of Study Subjects

In this study, 1196 visitors who responded from urban forests, national parks, and natural recreation forests were taken as a sample group. The gender was 613 males (51.3%) and 583 females (48.7%). The time it took to reach the forest was less than an hour for 896 visitors (74.9%), and 555 visitors (46.4%) spent 1 to 3 h inside the forests. Hiking/walking (60.4%) was the most popular reason for visiting, and the main reasons for visiting the forests related to COVID-19 were hiking 26.1%, exercise 22.2%, and walking 17.8%. Other general characteristics are as presented in Table 2.

### 3.2. Descriptive Statistics Analysis

The mean, standard deviation, skewness, and kurtosis were calculated. The results are presented in Table 3 to confirm the general tendency and normality of the perceived restorativeness and social–psychological stress, which are the main variables of this study. The mean of the perceived restorativeness was 5.31 (seven-point scale), and the sub-factors of the perceived restorativeness were “Diversion mood” 5.49, “Not boring” 5.58, “Coherence” 5.06, and “Compatibility” 5.22. The mean of social–psychological stress was 1.02 (three-point scale). Among the sub-factors of perceived restorativeness, it was found that the highest perception was not being bored. The skewness range of the variable was −0.95 to 0.22, and the kurtosis range was 0.04–1.67. Since the absolute value of skewness is less than 3 and kurtosis is less than 10, it can be seen that all variables satisfy the normal distribution assumption [36].

### 3.3. Correlation Analysis between Major Variables

Pearson correlation analysis was performed to confirm the association between the perceived restorativeness and social–psychological stress, which are the main variables of this study (Table 4). As a result of the analysis, it was found that the perceived restorativeness had a negative correlation with social–psychological stress (r = −0.40, *p* < 0.001). In other words, it was confirmed that the higher the perceived restorativeness, the lower the social–psychological stress.

### 3.4. Differences in Social–Psychological Stress Levels according to Demographic Characteristics

According to demographic characteristics, the chi-square test was conducted to verify the difference in social–psychological stress levels. As a result, differences in social–psychological stress levels according to age (χ2 = 20.50, *p* < 0.05), transportation used to visit the forest (χ2 = 19.71, *p* < 0.05), and time spent visiting the forest (χ2 = 16.82, *p* = 0.032) was found to be significant.

By age, the health group (*n* = 34, 35.8%) and the potential stress group (*n* = 318, 32.2%) showed a high level in their 50s, while the high-risk group (*n* = 34, 29.8%) showed a high level in their 60s. Through transportation, the health group (*n* = 35, 36.8%) and high-risk group (*n* = 41, 36.0%) showed the highest level in their on-foot, while the own car represented a high level in potential stress groups (*n* = 363, 36.8%). Finally, for residence time in the forest, the health group (*n* = 42, 44.2%), the potential stress group (*n* = 461, 46.7%), and the high-risk group (*n* = 52 and *n* = 45.6%) all showed a high level in stayed in the forest for 1 to 3 h.

On the other hand, differences according to gender, frequency of usual forest visits, time spent visiting forests, and companions to forest visits were found to be insignificant (Table 5).

### 3.5. Verification of Differences in Major Variables According to Forest Space Types

One-way ANOVA was conducted to verify the difference in the perceived restorativeness and social–psychological stress according to the forest space type. As can be seen in Table 6, there was a significant difference in overall perceived restorativeness according to forest space type (F = 9.87, *p* < 0.001). Furthermore, the results of ANOVA indicate that there was a significant difference in three subfactors of the PRS, except for one subscale, entitled “Compatibility”: “Diversion mood” (F = 4.83, *p* < 0.01), “Not boring” (F = 12.54, *p* < 0.001), and “Coherence” (F = 18.38, *p* < 0.001).

The analysis of the post hoc Scheffé test revealed that National parks and natural recreational forests were higher than urban forests in “Diversion mood”, a sub-factor of the PRS. Additionally, national parks were higher than urban forests in “Not bored”, and natural recreational forests were higher than urban forests in “Coherence”.

On the other hand, there is no significant difference in the social–psychological stress according to the forest space type.

## 4. Discussion

From 2020, to prevent the infection and spread of COVID-19, restrictions on daily life such as wearing masks, social distancing, living at home, and banning group gatherings have been prolonged. On the other hand, although the National Parks Authority suggested using some facilities, some national parks have seen increases in visitors compared to last year. In addition, visitors to forests near the city, which are easily accessible by private vehicles, are also increasing [28].

In this study, a face-to-face questionnaire survey was conducted on people who visited urban forests, national parks, and natural recreation forests from May 1 to July 15, 2020, while wearing a mask and complying with the government’s COVID-19 distancing policy to find out the rehabilitation environment perception and social–psychological stress of these forest visitors. In the 2020 Seoul citizens’ COVID-19 risk perception survey, 14.9% were found to be in a “highly stressed state requiring immediate help”. However, the social–psychological stress levels of the subjects (*n* = 1196) of this study were 8% in the “health group”, 82.5% in the “potential stress group”, and 9.5% in the “high-stress group”. The fact that the “high-risk group” of forest visitors is 10% or less means that forests can be a place to reduce stress, as studies show [14,16,20] that stress is relieved in a forest environment. In addition, as a result of Pearson correlation analysis, as in previous studies [37], the higher the perceived restorativeness, the lower the social–psychological stress was found to have a negative correlation. In this study, when “high-stress groups” stayed in the forest for less than 1 h, results such as χ2 = 16.82, *p* = 0.032 were obtained, indicating that people with high stress should stay in the forest 1 h or more to be effective. As Lam [38] mentioned that women have mental health problems and therefore have higher stress than men, this study also showed that women (56%) had a higher “high risk of stress” than men (44%). According to studies showing that the prevalence of anxiety, depression, and stress due to the spread of COVID-19 is higher in women than in men [39], and female college students experience more negative emotions than male college students [40], it can be seen that women are more stressed about COVID-19 than men.

This study showed that hiking/walking was the most popular reason for visits to urban forests, natural recreation parks, and national parks. Many previous studies have shown that forest walking or forest exercise provides a wide range of physiological and psychological health. For example, forest walking can suppress sympathetic nervous activity and increase parasympathetic nervous activity [41,42] and natural killer cell activity [43,44], and reduce cortisol concentration [45,46] and blood pressure [47,48]. In addition, regarding psychological health aspects, the effect of forest walking or exercise was associated with improved mood state [49,50], self-esteem [51], decreased stress, depression, and anxiety [52]. Therefore, due to the health mentioned above, visiting forests for “mountaineering/walking” can be an excellent motivation for visitors.

This result showed that perceived restorativeness and other subscales of the PRS such as “Diversion mood,” “Not boring,” and “Coherence” were different from the three forest types. Among them, the perceived restorativeness of recreation forests and national parks was higher than urban forests. It is thought that the fact that national parks and recreational forests are farther away from cities than urban forests contributed to the sense of recovery significantly. According to attention restoration theory (ART), the term “restoration” refers to people’s processes when recovering from something that has reduced their ability to cope with their everyday life tasks and demands [17]. One of these restoration environment elements is being-away, referring to a change of scenery and or separate from daily routines, promoting a conceptual distance from the ordinary, such as stressful situations in their living environments [21]. Significantly, the natural recreational forest recovery environment perception (5.40 ± 0.69) in this study was the highest. The natural recreational forest, designed to promote national health, is located 1 to 2 h away from the city center. Significant results in the sub-factors of “Diversion mood” and “Coherence” could be seen as affecting satisfaction in previous studies [53]. As a sub-factor of the national park’s perceived restorativeness (5.36 ± 0.79), it was high in “Not bored.” It has many natural features like previous studies [54], such as the size of the mountain, ecological value, and natural landscape.

However, this result showed that social–psychological stress was not different in the three forest types. These results mean that the forest environment may be effective regardless of forest space in relieving stress. Moreover, medical research has grown on the specific role of contact with forests in reducing stress levels [55,56,57,58,59]. For example, a recent systematic review of 43 studies found that heart rate, blood pressure, and self-report measures provide the most persuasive evidence that spending time in outdoor environments, particularly those with green space, may reduce the experience of stress and ultimately improve health [58]. A systematic review of 22 studies and 8 meta-analysis studies showed that exposure to the forest could significantly affect cortisol levels in such a way as to reduce stress [59]. In Ulrich’s [60] Stress Reduction Theory (SRT), positive responses and attention to staying in the natural environment or seeing natural elements could experience stress reduction, and it can be seen that the stress reduction did not depend on the forest space type. Resilience improvement to prevent stress and mental disorders resulted, as in Lee and Hyun [61], which showed the possibility that the recovery environment can exert a relief effect on not only stress but also life events [24], and it can be seen that the forest is provided as a recovery space to cope with stress.

This study is a face-to-face study conducted with forest visitors during a period of high national stress due to anxiety about the transmission of COVID-19. However, during the COVID-19 period, there was a limit to representing the whole as they practiced and proceeded with the distance in life in a limited place called a forest. In addition, as in previous studies, there was no significant difference between forest types with stressful natural landscapes in this study, but additional research is needed.

We believe that as adverse effects such as stress, anxiety, depression, etc., on COVID-19 infection and confirmation can cause severe mental problems [39], we believe visiting a nearby forest will be helpful for stress reduction [60] and health promotion [17]. In particular, it is reported that exercise in the forest can increase alpha and beta waves, improve brain health, and lead to a healthy life [62]. Therefore, for visitors who visit the forest to hike, exercise, and walk while being restricted in their daily life due to COVID-19, we should find ways to manage forests well and develop forest healing programs so that more people can visit forests and help promote health.

## 5. Conclusions

This study focuses on the effect of the perceived restorativeness of citizens visiting the forest during COVID-19 on social–psychological stress and the difference of significant variables according to the types of forest spaces. The results are as follows. (1) The higher the perceived restorativeness in the forest space, the lower the social–psychological stress. (2) The social–psychological stress in the forest space was found to be lower than that of the daily living space. (3) Depending on the type of forest space in Korea, there is a difference in variables in the sub-factors of the perceived restorativeness, such as “Diversion mood,” “Not boring,” and “Coherence.” (4) “Suitability,” a sub-factor of the perceived restorativeness in forests, is not limited by the type of forest space. (5) The social–psychological stress is not limited by the type of forest space. In conclusion, for citizens who visited the forest during the COVID-19 period, the perceived restorativeness about the forest space reduced social–psychological stress.

## Figures and Tables

**Figure 1 ijerph-18-06328-f001:**
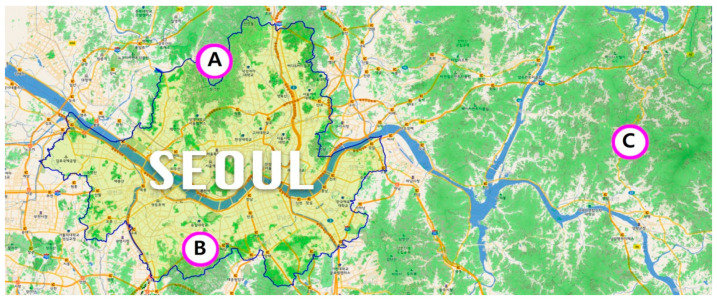
Research sites located in the suburbs of Seoul. (**A**) Mt. Gwanak Urban Forest; (**B**) Mt. Bukhan National Park; (**C**) Mt. Yumyeong Recreation Forest (https://map.forest.go.kr/forest/#/) (accessed on 7 June 2021).

**Table 1 ijerph-18-06328-t001:** Study area.

Forest Space Types	Urban Forest	National Park	Recreation Forest
Law	The term “Urban forest” refers to forests and trees created and managed in a city to promote citizens’ relaxation and health, emotional development, and experiential activities. Park areas under Article 2 of the “Nature Park Act” are excluded. (Act 2.1 on the Creation and Management of Urban Forests, etc.)	The term “national parks” means parks as regions worthy of representing the natural ecosystem, nature, and cultural scenery of the Republic of Korea, designated under Articles 4 and 4-2; (Natural Parks Act, 2.2]	The term “natural recreation forest” means a forest developed for citizens’ emotional development, relaxation for health, and education on forestry (including recreational facilities and the land therein) (Forestry Culture and Recreation Act 2.2]
Site	(A) Mt. Gwanak Urban Forest (Altitude: 629 m)	(B) Mt. Bukhan National Park (Altitude: 837 m)	(C) Mt. Yumyeong Recreation Forest (Altitude: 862 m)
Location	Gwanak-gu, Seoul, and Anyang City, Gwacheon City 126°56′52′′ E/37°27′49′′ N 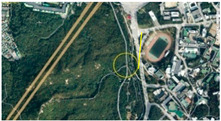	Gangbuk-gu, Seoul, and Goyang City 126°57′43′′ E/37°39′09′′ N 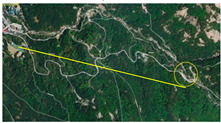	Gapyeong-Gun, Yangpyeong-Gun, Gyeonggi-do 127°49′18′′ E/37°58′83′′ N 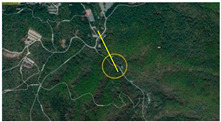
Vegetation	*Pinues densiflora*, *Alnus hirsuta*, *Quercus*, *Fraxinus*, *Fallopia convolvulus*, *Buxus*	*P. densiflora*, *A. hirsute, Quercus mongolica*, *Juniperus chinensis*, *Actinidia polygama*, *P. bungeana*	*Q. mongolica*, *Acer palmatum*, *Styrax obassia*, *Pinus koraiensis*, *Larix leptolepis*
Trail	Depending on the starting point of the trail, Gwanak-gu starting point (21), Gwacheon starting point (7), Anyang starting point (4), a total of 32.	A total of 97 sections, 217.57 km of trails were created.	Parking lot → Promenade fork → Mountain top → Madangso → Yongso → Batjaso → Valley → 6.96 km roundabout course returning to the parking lot.
100 famous mountains (Korea Forest Service)	**10th famous mountain****(Mt. Gwanak)** It has been selected as one of the five wonders of Gyeonggi since ancient times, with beautiful scenery and an urban nature park (designated in 1968) located close to the downtown area, considering that it is a resting place for the residents of the metropolitan area.	**47th famous mountain (Mt. Bukhan)** It has been selected considering the beautiful scenery such as Insubong, Mangyongdae, and Nojeokbong, including Baekundae, the highest peak, and a resting place for city dwellers and designated as a national park (in 1983).	**68th famous mountain****(Mt. Yumyeong Recreation Forest)** It has been selected considering the gentle and soft ridgeline, the valley with abundant water, and the beautiful scenery in harmony with strange rocks and lush forests.
COVID-19 Restriction on Using Forest	There was no restriction.	Promotion of safety rules such as 2 m social distancing, wearing a mask, and walking in one line to the right for incoming visitors.	Natural recreation forest facilities that were reserved online were completely stopped, and hiking trails were opened for a fee.
Survey site photo	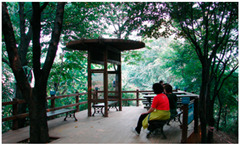 https://blog.daum.net/sansol/1605 (accessed on 7 June 2021)	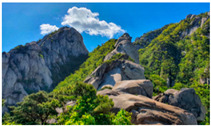 https://www.facebook.com/iloveknp/photos/pcb.4014255608653150/4014255101986534/ (accessed on 7 June 2021)	* 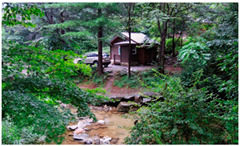 * https://m.blog.daum.net/sanpool/15959015 (accessed on 7 June 2021)

**Table 2 ijerph-18-06328-t002:** Descriptive characteristics of the study subjects.

Variable		Urban Forest	National Park	Natural Recreation Forest	Total
Sex	Male	176 (45.7)	281 (57.0)	156 (49.1)	613 (51.3)
	Female	209 (54.3)	212 (43.0)	162 (50.9)	583 (48.7)
Age	Less than 30	37 (9.6)	26 (5.3)	23 (7.2)	86 (7.2)
	30–39	23 (6.0)	51 (10.3)	27 (8.5)	101 (8.4)
	40–49	44 (11.4)	65 (13.2)	83 (26.1)	192 (16.1)
	50–59	103 (26.8)	163 (33.1)	114 (35.8)	380 (31.8)
	60–69	129 (33.5)	162 (32.9)	63 (19.8)	354 (29.6)
	More than 70	49 (12.7)	26 (5.3)	8 (2.5)	83 (6.9)
Forest visit frequency	almost everyday	42 (10.9)	21 (4.3)	13 (4.1)	76 (6.4)
	1–2 times/week	180 (46.8)	250 (50.7)	94 (29.6)	524 (43.8)
	1–2 times/month	89 (23.1)	143 (29.0)	107 (33.6)	339 (28.3)
	1–2 times/6 months	35 (9.1)	36 (7.3)	63 (19.8)	134 (11.2)
	1–2 times/year	19 (4.9)	32 (6.5)	34 (10.7)	85 (7.1)
	Almost never	20 (5.2)	11 (2.2)	7 (2.2)	38 (3.2)
Transportation when visiting the forest	On foot	182 (47.3)	129 (26.2)	89 (28.0)	400 (33.4)
	Bicycle	6 (1.6)	14 (2.8)	4 (1.3)	24 (2.0)
	Own car	50 (13.0)	196 (39.8)	183 (57.5)	429 (35.9)
	Taxi	4 (1.0)	5 (1.0)	1 (0.3)	10 (0.8)
	Subway	87 (22.6)	133 (27.0)	28 (8.8)	248 (20.7)
	Other	56 (14.5)	16 (3.2)	13 (4.1)	85 (7.1)
Time to reach the forest	Less than 10 min	47 (12.2)	48 (9.7)	39 (12.3)	134 (11.2)
	10–30 min	112 (29.1)	137 (27.8)	78 (24.5)	327 (27.3)
	0.5–1 h	149 (38.7)	200 (40.6)	86 (27.0)	435 (36.4)
	1–2 h	56 (14.5)	90 (18.3)	75 (23.6)	221 (18.5)
	2–5 h	18 (4.7)	14 (2.8)	39 (12.3)	71 (5.9)
	More than 5 h	3 (0.8)	4 (0.8)	1 (0.3)	8 (0.7)
People coming together into the forest	Alone	109 (28.3)	127 (25.8)	48 (15.1)	284 (23.7)
	Friend	142 (36.9)	133 (27.0)	88 (27.7)	363 (30.4)
	Colleague	15 (3.9)	22 (4.5)	13 (4.1)	50 (4.2)
	Family	115 (29.9)	202 (41.0)	164 (51.6)	481 (40.2)
	Other	4 (1.0)	9 (1.8)	5 (1.6)	18 (1.5)
Time staying in the forest	Less than 30 min	18 (4.7)	17 (3.4)	7 (2.2)	42 (3.5)
	0.5–1 h	86 (22.3)	67 (13.6)	53 (16.7)	206 (17.2)
	1–3 h	189 (49.1)	199 (40.4)	167 (52.5)	555 (46.4)
	3–5 h	82 (21.3)	170 (34.5)	73 (23.0)	325 (27.2)
	More than 5 h	10 (2.6)	40 (8.1)	18 (5.7)	68 (5.7)
Activities in the forest (duplicate response)	Mountaineering/walking	331 (57.3)	429 (63.0)	283 (60.3)	1043 (60.4)
	Visiting cultural properties	61 (10.6)	77 (11.3)	68 (14.5)	206 (11.9)
	Cultural property viewing	5 (0.9)	10 (1.5)	10 (2.1)	25 (1.4)
	Relaxation/meditation	115 (19.9)	106 (15.6)	67 (14.3)	288 (16.7)
	Festival event	6 (1.0)	1 (0.1)	3 (0.6)	10 (0.6)
	Photo shoot	22 (3.8)	30 (4.4)	23 (4.9)	75 (4.3)
	Use of mineral water	8 (1.4)	7 (1.0)	2 (0.4)	17 (1.0)
	Use of sports facilities	24 (4.2)	11 (1.6)	8 (1.7)	43 (2.5)
	Environment commentary	3 (0.5)	3 (0.4)	1 (0.2)	7 (0.4)
	Other	3 (0.5)	7 (1.0)	4 (0.9)	14 (0.8)
Advantages of visiting the forest (duplicate response)	Fresh air	266 (35.2)	303 (30.3)	231 (31.4)	803 (32.1)
	Nature sounds and tranquility	125 (16.5)	179 (17.7)	168 (22.8)	472 (18.9)
	Beautiful scenery	88 (11.6)	165 (16.3)	113 (15.4)	366 (14.6)
	Scent of nature	123 (16.3)	147 (14.6)	106 (14.4)	376 (15.0)
	Refreshing from activity	152 (20.1)	207 (20.5)	113 (15.4)	472 (18.9)
	Other	2 (0.3)	6 (0.6)	5 (0.7)	13 (0.5)
Reasons to visit the forest in connection with COVID-19	Boost immunity	27 (7.0)	17 (3.4)	3 (0.9)	47 (3.9)
	Relieve stress	18 (4.7)	26 (5.3)	30 (9.4)	74 (6.2)
	Exercise	99 (25.7)	144 (29.2)	23 (7.2)	266 (22.2)
	Walk	107 (27.8)	29 (5.9)	77 (24.2)	213 (17.8)
	Fresh air	40 (10.4)	27 (5.5)	25 (7.9)	92 (7.7)
	Mountain climbing	52 (13.5)	207 (42.0)	53 (16.7)	312 (26.1)
	Rest	28 (7.3)	33 (6.7)	66 (20.8)	127 (10.6)
	Other	14 (3.6)	10 (2.0)	41 (12.9)	65 (5.4)

**Table 3 ijerph-18-06328-t003:** Technical statistics of perceived restorativeness and social∙psychological stress variables.

Variable	Mean	SD	Min	Max	Skew	Kurtosis
Perceived restorativeness	5.31	0.77	2.69	7.00	−0.23	−0.08
Diversion mood	5.49	0.99	1.00	7.00	−0.95	1.67
Not boring	5.58	1.17	1.00	7.00	−0.68	0.04
Coherence	5.06	0.92	1.50	7.00	−0.22	0.12
Compatibility	5.22	1.07	1.00	7.00	−0.57	0.56
Social–psychological stress	1.02	0.37	0	2.67	−0.29	0.49

Note: SD, standard deviation; Min, minimum; Max, maximum.

**Table 4 ijerph-18-06328-t004:** Correlation analysis between the perceived restorativeness and social∙psychological stress variables.

Variable	Perceived Restorativeness	Social–Psychological Stress
Perceived Restorativeness	1	
Social–psychological stress	−0.40 ***	1

*** *p* < 0.001.

**Table 5 ijerph-18-06328-t005:** Differences in social–psychological stress levels according to demographic characteristics.

Variable		Health Group (*n* = 95)	Potential Stress Group (*n* = 987)	High Risk Group (*n* = 114)		
Sex	Male	50 (52.6)	513 (52.0)	50 (43.9)	χ²	2.770
	Female	45 (47.4)	474 (48.0)	64 (56.1)	*p*	0.250
Age	Less than 30	12 (12.6)	60 (6.1)	14 (12.3)	χ²	20.50
	30–39	8 (8.4)	86 (8.7)	7 (6.1)	*p*	0.025 *
	40–49	15 (15.8)	151 (15.3)	26 (22.8)		
	50–59	34 (35.8)	318 (32.2)	28 (24.6)		
	60–69	22 (23.2)	298 (30.2)	34 (29.8)		
	More than 70	4 (4.2)	74 (7.5)	5 (4.4)		
Forest visit frequency	Almost everyday	8 (8.4)	63 (6.4)	5 (4.4)	χ²	9.090
	1–2 times/week	45 (47.4)	437 (44.3)	42 (36.8)	*p*	0.524
	1–2 times/month	20 (21.1)	285 (28.9)	34 (29.8)		
	1–2 times/6 months	11 (11.6)	107 (10.8)	16 (14.0)		
	1–2 times/year	8 (8.4)	66 (6.7)	11 (9.6)		
	Almost never	3 (3.2)	29 (2.9)	6 (5.3)		
Transportation when visiting the forest	On foot	35 (36.8)	324 (32.8)	41 (36.0)	χ²	19.710
	Bicycle	0 (0)	17 (1.7)	7 (6.1)	*p*	0.032 *
	Own car	33 (34.7)	363 (36.8)	33 (28.9)		
	Taxi	1 (1.1)	8 (0.8)	1 (0.9)		
	Subway	23 (24.2)	198 (20.1)	27 (23.7)		
	Other	3 (3.2)	77 (7.8)	5 (4.4)		
Time to reach the forest	Less than 10 min	15 (15.8)	107 (10.8)	12 (10.5)	χ²	10.830
	10–30 min	27 (28.4)	271 (27.5)	29 (25.4)	*p*	0.371
	0.5–1 h	34 (35.8)	365 (37.0)	36 (31.6)		
	1–2 h	15 (18.8)	183 (18.5)	23 (20.2)		
	2–5 h	4 (4.2)	54 (5.5)	13 (11.4)		
	More than 5 h	0 (0)	7 (0.7)	1 (0.9)		
People coming together into the forest	Alone	20 (21.1)	236 (23.9)	28 (24.6)	χ²	7.120
	Friend	20 (21.1)	308 (31.2)	35 (30.7)	*p*	0.523
	Colleague	5 (5.3)	39 (4.0)	6 (5.3)		
	Family	48 (50.5)	390 (39.5)	43 (37.7)		
	Other	2 (2.1)	14 (1.4)	2 (1.8)		
Time to stay in the forest	Less than 30 min	3 (3.2)	32 (3.2)	7 (6.1)	χ²	16.820
	0.5–1 h	10 (10.5)	171 (17.3)	25 (21.9)	*p*	0.032 *
	1–3 h	42 (44.2)	461 (46.7)	52 (45.6)		
	3–5 h	28 (29.5)	272 (27.6)	25 (21.9)		
	More than 5 h	12 (12.6)	51 (5.2)	5 (4.4)		

Note: 0–8 points were defined as a healthy group; 9–26 points, a potential stress group; and 27–54 points were defined as a high-risk group, * *p*< 0.05.

**Table 6 ijerph-18-06328-t006:** Verification of major variable differences according to forest type.

Variable	Group	N	M ± SD	F	Scheffé
PRS	Urban Forest (a)	385	5.17 ± 0.77	9.87 *** (0.000)	b,c > a
	National Park (b)	493	5.36 ± 0.79		
	Natural recreation forest (c)	318	5.40 ± 0.69		
Diversion mood	Urban Forest (a)	385	5.36 ± 0.97	4.83 ** (0.008)	b,c > a
	National Park (b)	493	5.54 ± 1.06		
	Natural recreation forest (c)	318	5.57 ± 0.91		
Not boring	Urban Forest (a)	385	5.36 ± 1.25	12.54 *** (0.000)	b > a
	National Park (b)	493	5.76 ± 1.12		
	Natural recreation forest (c)	318	5.57 ± 1.10		
Coherence	Urban Forest (a)	385	4.89 ± 0.97	18.38 *** (0.000)	c > a,b
	National Park (b)	493	5.03 ± 0.95		
	Natural recreation forest (c)	318	5.31 ± 0.77		
Compatibility	Urban Forest (a)	385	5.12 ± 1.09	2.59 (0.076)	-
	National Park (b)	493	5.28 ± 1.12		
	Natural recreation forest (c)	318	5.24 ± 0.96		
Social–psychological stress	Urban Forest (a)	385	1.05 ± 0.35	2.82 (0.060)	-
	National Park (b)	493	0.99 ± 0.41		
	Natural recreation forest (c)	318	1.04 ± 0.34		

Note: ** *p* < 0.01, *** *p* < 0.001.

## Data Availability

The date presented in this study are available on request from the corresponding author. The data are not publicly available due to privacy.

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
