# Peer review of "Influence of Forest Visitors’ Perceived Restorativeness on Social–Psychological Stress"

_ijerph, 2021, doi:10.3390/ijerph18126328_

Round 1

Reviewer 1 Report

This is an interesting article that analyzes the psychological resilience of forest visitors in different space types of forests.

But some parts must be improved with more explanations and discussions.

  1.  

In order to take a survey and publish the results as a paper, you need the approval of the Ethics Committee. This paper does not state this.

2.

The conditions of the subjects are insufficient. Did you include small children? Did all people in the same family or group respond? If three people in the same family responded, three people will respond with the same means of transportation and time to reach the forest. If 10 people in the same group of friends respond, 10 people will give the same answer for time to reach the forest. Doesn't this lead to uneven results?

3.

Line 129-. "A total of 1,196 questionnaires were adopted as data for this study, excluding 5 unfaithful questionnaires among a total of 1,211 questionnaires. A total of 1,196 questionnaires were adopted as data for this study, excluding 5 unfaithful questionnaires among a total of 1,211 questionnaires.

→”5 unfaithful questionnaires”.

Please provide a brief description of what is unfaithful.

  1.  

Please describe the details of the “Urban Forest”, “National Park” and “Natural Recreation Forest” in terms of area, height difference, forest coverage rate, and facilities (restaurants, BBQ space, playground equipment, rest areas, etc.)?

  1.  

Table 3&4. Are these the results of the questionnaires for all subjects? Please add an explanation.

6.

Table5. This shows the percentages of the Health group, Potential group, and High risk group for each item. However, while there may be a high percentage of people in the high-risk group under 30 and in their 40s, the largest number of people in the high risk group are in their 60s, and the number of people in the high risk group of 34 is not negligible. The same is true for time spent in the forest, with the high-risk group actually having more people who spent 1-3 hours in the forest. The fact that this is not mentioned makes it seem that only the most convenient data was extracted. The aggregate data for each group seems to be more important.

7.

Table6. In almost all items, the score for “Natural recreation forest (c)” is high. Please discuss which factor is leading to such a result.

8.

The author mentions a lot about the effect of relaxing in the forest, but the results of the questionnaire shows that the participants emphasize walking, hiking, and exercising more than relaxing. I think the author needs to discuss the benefits of exercising in the forest.

Author Response

Dear reviewer,

We would like to express our sincere gratitude for your kind consideration and comments on our manuscript. According to reviewers’ comments and suggestion, we revised the manuscript as follows:

Reviewer #1:

  1. It has been confirmed that this study is subject to deliberation exemption in accordance with Article 16 of the Bioethics Act and Article 13 of the Enforcement Regulations were as follows (line 361-369).
  2. We inserted contents that a questionnaire was conducted with 19-year-old adults who visited the forest (lines 118-123).
  3. We inserted contents that questionnaires that did not mark all of the questionnaire items were excluded (line 123-125).
  4. We inserted that describe the details of the “Urban Forest”, “National Park” and “Natural Recreation Forest” in Table 1 (line 133-156).
  5. We inserted an explanation about Table 3 and Table 4 (line 205-208).
  6. We analyzed again and inserted content about the important results between each group (line 225-237; 242-254).
  7. We added a discussion on why the results of natural recreation forests are high in most items (line 294-311).
  8. We inserted a discussion on the benefits of exercising in the forest (line 284-293).

Reviewer 2 Report

This paper is very good timing one in present COVID-19 situation!

Especially, this survey analyzed big data (approximately 1200 people) and deal with three type forests. The points of the time and transportation to reach the forest are fine, too. These results must give a big impact on forest studies!

However, if you can add, please consider matters more as below;

  1. You studied three forest types (urban forest, national park, and recreation forest). But how about the geographical and vegetational differences of them? They are important points of views for walkers and visitors. If you can, please describe the forest conditions.  
  2. The results showed that there are not so big differences among the three forest types. How do you think about this point? Please add your comment.  

Author Response

Dear reviewer,

We would like to express our sincere gratitude for your kind consideration and comments on our manuscript. According to reviewers’ comments and suggestion, we revised the manuscript as follows:

Reviewer #2:

  1. We inserted the geographical and vegetational differences of three forest therapy (urban forest, national park, and recreation forest) in Table 1 and Figure 1 (lines 133-156).
  2. We inserted differences in forest space types for perceived resilience and social and psychological stress in the discussion (line 294-333).

Reviewer 3 Report

Dear Authors,

thank you very much for submitting this very interesting, very well conducted study. I suggest publishing the work after some minor revisions.

My main point would be including some more literature in the discussion section as there have been some studies in other regions of the world. It is not an absolute must but it would be worthwile to discuss your findings in such a broader context involving some of the exisiting literature on health benefits of forests or green space as it would significantly enhance your paper.

Literature for this would be something like Simkin et al. (2020): Restorative effects of mature and young commercial forests, pristine old-growth forest and urban recreation forest - A field experiment, Urban Forestry & Urban Greening or if you pick up some more general literature for green space it could be some literature like Twohig-Bennett & Jones (2017): The health benefits of the great outdoors: A systematic review and meta-analysis of greenspace exposure and health outcomes, Environmental Research or Markevych et al. (2017): Exploring pathways linking greenspace to health: Theoretical and methodological guidance, Environmental Research.

Having read those papers recently, they came up to my mind when reading your discussion section. They are just examples and it is fine if you pefer or opt for other, more suiting papers for discussing your findings.

Other small comments:

Methods section: A simple map showing the location of the different forests would be very nice for the reader to get some idea about the study area.

Line 70-82: A citation or reference would be good for these statements.

Line 292: Sentence seems to be incomplete.

Author Response

Dear reviewer,

We would like to express our sincere gratitude for your kind consideration and comments on our manuscript. According to reviewers’ comments and suggestion, we revised the manuscript as follows:

Reviewer #3:

  1. We inserted a discussion on more literature about the health benefits of forest or greenspace (line 284-327).
  2. We inserted that a simple map showing the location of the different forests in Table 1 and Figure 1 (line 153-154).
  3. We modified the contents of ART theory (line 67-73).
  4. We revised it to a complete sentence (line 328-329).